# Safety and Feasibility of Single-Port Trans-Axillary Robotic Thyroidectomy: Experience through Consecutive 100 Cases

**DOI:** 10.3390/medicina58101486

**Published:** 2022-10-19

**Authors:** Il Ku Kang, Joonseon Park, Ja Seong Bae, Jeong Soo Kim, Kwangsoon Kim

**Affiliations:** Department of Surgery, College of Medicine, The Catholic University of Korea, Seoul 06591, Korea

**Keywords:** robotic thyroidectomy, single-port robotic system, safety and feasibility, trans-axillary thyroidectomy

## Abstract

*Background and Objectives:* Recently, the single-port (SP) robotic system was introduced for minimally invasive operative techniques. Thus, this study aimed to validate the safety and feasibility of SP trans-axillary robotic thyroidectomy (SP-TART) through experiences in a single tertiary institution. *Materials and Methods:* This study retrospectively analyzed 100 consecutive patients who underwent SP-TART from October 2021 to June 2022 in Seoul St. Mary’s Hospital in Seoul, Korea. We analyzed the clinicopathological characteristics and perioperative outcomes, including complications. *Results:* Less than total thyroidectomy (LTT) was performed in 81, total thyroidectomy (TT) in 16, and TT with modified radical neck dissection (mRND) in 3 patients. The mean operation time (min) was 53.3 ± 13.7, 86.3 ± 15.1, and 245.7 ± 36.7 in LTT, TT, and TT with mRND, respectively. The mean postoperative hospital stay was 2.0 ± 0.2, 2.1 ± 0.3, and 3.7 ± 1.5 days, respectively. A total of 84 cases of thyroid cancer were included, and 97.6% of them (82 cases) were papillary carcinoma and the rest were follicular and poorly differentiated carcinomas. Regarding complications, five cases had major complications, including three cases of vocal cord palsy and two cases of transient hypoparathyroidism. *Conclusions:* SP-TART is safe and feasible with a short operation time and a short length of hospital stay.

## 1. Introduction

Thyroid cancer is the most common cause of thyroidectomy. An estimated 43,800 adults will be diagnosed with thyroid cancer this year in the United States according to cancer statistics released by the American Cancer Society [1].

In the mid-2000s, robotic thyroidectomy was first introduced using a gasless trans-axillary approach [2,3]. Then, endocrine surgeons tried various types of remote access approaches, such as the bilateral axillo-breast, the gasless post-auricular facelift, and the trans-oral approach [4,5]. Trans-axillary robotic thyroidectomy (TART) is an application of an endoscopic thyroidectomy to a robotic system. Huscher et al. performed the first endoscopic thyroidectomy after Gagner pioneered the endoscopic parathyroidectomy [6,7]. Later, Chung et al. devised a gasless trans-axillary technique [8,9].

Conventional open thyroidectomy (COT) using Kocher’s incision was long considered a safe operation among surgeons. However, COT is associated with two disadvantages. One is swallowing discomfort due to adhesion between the subplatysmal flap and the strap muscles [10]. Two is the scar location which is prone to exposure [11]. TART with a multi-port robotic system became widespread in Asian countries, where people care about other people’s eyes or opinions because it can hide scars. Additionally, the short access distance from the armpit of Asian patients would have helped surgeons try TART in the early stage.

TART with the previous robotic system still has drawbacks [8,12], such as longer incisions and a broader extent of the skin flap than COT. Endoscopic thyroidectomy uses the same surgical instruments as in laparoscopic surgery, whereas TART uses three robotic arms with wrist articulation for counter-traction, dissection, and cutting with cauterization. However, the two methods require the same anatomical working space where an external retractor is inserted to lift the sternal head of the sternocleidomastoid muscle (SCM) and strap muscles surrounding the thyroid gland [2]. Therefore, some patients have postoperative pain and paresthesia in the supra- and infra-clavicular areas [13].

The single-port (SP) robotic system (Intuitive Surgical Inc., Sunnyvale, CA, USA) has advantages when it comes to thyroid surgery. SP trans-axillary robotic thyroidectomy (SP-TART) requires a shorter incision length and a narrower skin flap for working space that enables less sensory nerve injury [5]. However, little is known about its safety and feasibility. Thus, this study aimed to share experiences and help endocrine surgeons who would like to introduce SP-TART by reporting consecutively performed 100 cases in a single tertiary institution.

## 2. Materials and Methods

### 2.1. Patients

This study retrospectively analyzed 100 consecutive patients who underwent SP-TART from October 2021 to June 2022 in Seoul St. Mary’s Hospital in Seoul, Korea. All patients who underwent SP-TART during the period were enrolled regardless of pathology or surgical extent. All operations were performed by a single surgeon (K.K.). The medical charts and pathology reports were reviewed and analyzed. There were no selection criteria for SP-TART, which was performed for all the patients who agreed with our procedure and possible complications.

This study was conducted following the Declaration of Helsinki (as revised in 2013) and approved by the Institutional Review Board of Seoul St. Mary’s Hospital, the Catholic University of Korea (approval No.: KC22RISI0657), which waived the requirement for informed consent due to the retrospective nature of the study.

### 2.2. Operative Procedure of Less Than Total Thyroidectomy (LTT)

The patient is positioned supine with a soft pillow behind their shoulders to allow for neck extension under general anesthesia. The lesion-side arm is lifted overhead and fastened to expose the axilla. Landmarks, such as the cricoid cartilage, the sternal notch, and the upper pole of the ipsilateral lobe, are marked. A 3.5–4 cm curvilinear skin incision is made just posterior to the anterior axillary line so that the scar is covered when the arm is lowered. A subcutaneous skin flap is made along the anterior surface of the pectoralis major muscle and clavicle until the SCM is identified. The SCM is exposed after opening the superficial layer of the deep cervical fascia and raising the subplatysmal skin flap. The avascular plane between the sternal and clavicular heads of the SCM is identified and separated. The dissection is continued until the omohyoid, and other strap muscles are identified. The ipsilateral lobe of the thyroid gland is exposed after lifting the lateral border of the strap musculature upward and the internal jugular vein (IJV) downward. The space between the strap muscles and the thyroid gland is further dissected. After visualizing the upper pole of the ipsilateral lobe and part of the contralateral lobe of the thyroid gland, an external retractor [2] is inserted through the axillary skin incision to elevate the strap muscles. The procedure described above is based on the gasless method without using CO_2_ insufflation during surgery.

The camera and instruments are inserted into the surgical field through a single cannula placed lateral to the skin incision. In the cannula, a camera is placed on the bottom; two pairs of Maryland dissecting forceps with bipolar electrocautery are positioned on both lateral-side arms, and a pair of Cadiere forceps are inserted on top.

All dissection and vessel ligations are performed via the Maryland bipolar dissectors. Intraoperative neuromonitoring is utilized for all patients. Thyroidectomy proceeded in the same manner as previous trans-axillary approaches with a single incision. The operator is seated at the console, and the assistant is placed next to the lesion-side of the patient. The clavicular head of the SCM and the IJV are softly retracted downward with the suction device by the assistant. The ipsilateral lobe of the thyroid gland is retracted in the caudal direction. Dissection proceeded until the upper pole of the ipsilateral lobe is sufficiently mobilized. The superior thyroidal vessels are ligated in contact with the thyroid gland to preserve the external branch of the superior laryngeal nerve. The superior and inferior parathyroid glands are identified, dissected, and gently swept laterally while keeping the vasculature intact. Cadiere forceps are used to medially retract the ipsilateral lobe to better visualize the tracheoesophageal groove and the course of the recurrent laryngeal nerve (RLN). The perithyroidal tissue is dissected keeping the nerve from thermal injury by the energy device once the nerve is identified near the lower pole of the ipsilateral lobe. The ipsilateral lobe is carefully retracted and dissected from the trachea using a Maryland dissector. Care is taken to avoid thermal injury to the RLN at the ligament of Berry. The surgical specimen is removed through the axillary skin incision with a laparoscopic grasper. The Nerve Integrity Monitor (Medtronic Inc., Minneapolis, MN, USA) is utilized to confirm the RLN function. After the closed suction drain insertion, the axillary skin incision is closed with an absorbable subcuticular skin stapler (INSORB, Incisive Surgical Inc., Plymouth, MN, USA).

### 2.3. Operative Procedure of Total Thyroidectomy (TT)

The flap dissection for working space and the ipsilateral thyroid lobectomy is the same as the LTT procedure. However, an external retractor is inserted in a slightly upper direction than ipsilateral lobectomy due to the difficulty of approaching the contralateral upper pole of the thyroid gland. The contralateral thyroid lobectomy proceeds after taking out the specimen of the ipsilateral lobe of the thyroid gland of the neck.

The isthmic portion is first dissected and detached from the trachea to complete the thyroidectomy. Afterward, the upper pole of the contralateral lobe is dissected to be sufficiently mobilized. Care should be taken for the course of the RLN and the location of the parathyroid glands when dissection proceeds.

### 2.4. Operative Procedure of Modified Radical Neck Dissection (mRND)

The patient is placed in the supine position on the operating table with a soft bar below the shoulders under general anesthesia. The lesion-side arm is abducted to expose the axilla and lateral neck. A 7–10-cm skin incision is made in the axilla along the anterior axillary fold and the lateral border of the pectoralis major muscle. The flap is medially dissected over the SCM toward the midline of the anterior neck under the platysma muscle. Laterally, the trapezius muscle is identified and dissected upward along its anterior border. The mRND procedure is similar to that of the open-conventional procedure after the docking procedure with an external retractor. After finishing levels III, IV, and V node dissections, redocking is required to provide a better view for level II lymph node (LN) dissection. The external retractor is removed and reinserted through the axillary incision toward the submandibular gland superiorly. A drainage tube is inserted under the incision line after delivering the specimen. Finally, the wound is cosmetically closed [14].

### 2.5. Surgical Outcome Assessment

Surgical outcomes were evaluated based on the medical chart review of all patients to collect data on tumor size, multifocality, extrathyroidal extension (ETE), thyroiditis, number of harvested and metastatic LNs, amount of blood loss, postoperative hospital stay, and postoperative complications. The pathologic stage was classified according to the 8th edition of the American Joint Committee on Cancer/Union for International Cancer Control (AJCC/UICC) tumor-node-metastasis (TNM) staging system. The duration of surgical phases, including flap elevation, docking, console operation, and total operation times, was determined.

## 3. Results

### 3.1. Baseline Clinicopathological Characteristics of the Study Patients

Table 1 presents the clinicopathological characteristics of 100 cases in the study. The mean age was 41.8 ± 13.3 (range, 17–69) years, and 9 (9.0%) patients were male. The mean body mass index was 23.4 ± 3.6 (range, 16.5–32.8) kg/m^2^. LTT was performed in 81 (81.0%) patients, which included lobectomy with tumerectomy, isthmusectomy, and simple lobectomy. One patient was confirmed to have a poorly differentiated thyroid carcinoma (PDTC) after lobectomy and underwent completion of TT. Additionally, 16 (16.0%) patients underwent TT, including this case, while TT with mRND was performed in 3 (3.0%) patients. In terms of pathology, papillary carcinoma (PTC) accounted for 82%, while follicular adenoma and noninvasive follicular neoplasm with papillary-like nuclear features were 8% and 3%, respectively. The mean tumor size was 1.5 ± 1.3 (range, 0.3–7.0) cm and 1.3 ± 1.2 (range, 0.1–5.9) cm, when measured in ultrasound and pathology, respectively. Thyroiditis was confirmed in 46 (46.0%) cases.

### 3.2. Perioperative Outcomes According to the Extent of Operation

Table 2 summarizes the perioperative outcomes according to the extent of the operation. the mean operation time (min) was 53.3 ± 13.7, 86.3 ± 15.1, and 245.7 ± 36.7, and the average amount of blood loss (ml) was 8.4 ± 5.4, 23.3 ± 48.9, and 38.3 ± 53.5 in LTT, TT, and TT with mRND, respectively. The mean postoperative hospital stay was 2.0 ± 0.2, 2.1 ± 0.3, and 3.7 ± 1.5 days, respectively. Out of 9 patients who had postoperative complications, 6 (7.4%) underwent LTT, 2 (12.5%) underwent TT, and 1 (33.3%) underwent TT with mRND.

### 3.3. Clinicopathological Characteristics of Patients with Cancer

The clinicopathological characteristics of patients with cancer are demonstrated in Table 3. On a pathological basis, 84 cases were cancer. Additionally, 82 (97.6%) cases were PTC, while the others were 1 (1.2%) follicular carcinoma and 1 (1.2%) PDTC. Surgical extent accounted for 79.8%, 20.2%, and 3.6% for LTT, TT, and TT with mRND, respectively. The mean tumor size was 1.3 ± 1.3 (range, 0.3–7.0) cm and 1.1 ± 1.2 (range, 0.1–5.9) cm, when measured in ultrasound and pathology, respectively. Multifocal tumors were observed in 30 (35.7%) cases. Tumors were invaded to the outside of the capsule of the thyroid gland in approximately two-thirds of the patients with cancer (55 cases, 65.5%), of which 5 (6.0%) cases had gross ETE. Thyroiditis was pathologically confirmed in half of the patients with cancer (42/84 cases). In terms of 81 patients (96.4% of cancer cases) who underwent LN dissection, the mean harvested LNs was 8.0 ± 9.0 (range, 0–56), while LNs positive for metastasis were 2.4 ± 4.8 (range, 0–27). According to the eighth version of the AJCC/UICC TNM staging system, 79 (94.0%) patients were pathological stage I, and 5 (6.0%) were stage II.

### 3.4. Postoperative Complications of the Study Participants

Postoperative complications are listed in Table 4. Nine patients experienced postoperative complications, including three vocal cord palsy, two surgical site infections, and two transient hypoparathyroidisms. Additionally, chyle leakage, bleeding at the drain insertion site, and wound dehiscence occurred in one case each. Of the patients with vocal cord palsy, two patients developed after LTT while one patient was after TT. A case of intraoperative transection of the recurrent laryngeal nerve was developed in one patient during left thyroid lobectomy.

## 4. Discussion

The concept of endocrine surgery has changed with time. In the past, surgeons were focused on the organ that had to be removed. Some patients suffered from voice change or hypocalcemia postoperatively. Nowadays, both surgeons and patients are interested in preserving the function of adjacent structures as much as possible. Since Gagner published a paper on the first endoscopic subtotal parathyroidectomy, many have developed remote-access thyroid surgery with various approaches [6,7]. However, the experience of TART is limited to a few countries, such as South Korea [15]. Concerns about different surgical outcomes between TART and COT remained [16]. We analyzed 100 consecutive patients who underwent SP-TART at a single tertiary institution. Herein, we would like to discuss our learning from our experience.

SP robotic system has a single arm with a flexible scope and three multi-jointed instruments that can easily approach blind spots without preparing a broad flap dissection [17,18]. The camera and instruments in the previous multi-port system have no flexibility, but it was multiple arms connected with them that moved. In comparison, the latest flexible system has instruments with multiple joints in the distal part to move like a fiber-optic endoscope. Making a long vertical incision and creating a broad flap dissection along the pectoralis major muscle and the clavicle was not necessary because the proximal part of the instruments and the camera are rigid and clustered like a bundle in a single cannula. The three-dimensional high-resolution camera is fully wristed to reach narrow spaces or blind spots. With the high resolution of the camera and the counter-traction of the third arm, differentiating the parathyroid glands and finding the course of the RLN was much easier in TART than in COT.

Kim et al. studied 200 cases of SP-TART and revealed that the position of the patient’s arm was laterally extended during upper pole dissection [18]. In contrast, we never had the patient’s arm moved during surgery. All patients were kept in the same position during all surgical procedures. The lesion-side arm was fixed overhead so that the armpit could be as close to the neck as possible for those who underwent LTT and TT. Patients who underwent TT with mRND were positioned with their lesion-side arm laterally extended.

A 3.5–4-cm long incision was made for LTT and TT, while a 7–8-cm long incision for TT with mRND. Based on our experience, relatively short incisions and a narrow extent of flap dissection were needed for SP-TART compared to the previous robotic system. A 3.5–4 cm long incision was sufficient to make a working space and for the instruments and camera to move, regardless of tumor size. Despite the smaller incision size, putting gauze in or taking the specimen out was not as difficult for the assistant as the previous multi-port robotic system. Less collision with the instruments occurred in the SP robot setting.

The operative view of TART is a lateral view which is similar to COT. Manipulating the upper pole of the ipsilateral lobe and dissecting the central LNs is relatively easier compared with the bilateral axillo-breast or trans-oral approach [2,19]. However, the upper pole of the contralateral lobe is more difficult to approach when performing TT. In this study, the console time was 22.5 ± 9.9 min in the LTT group (n = 81), while 47.8 ± 12.2 min in the TT group (n = 16), which was more than twice as long. This time difference will be shorter, as the surgeon gets more experienced.

The absence of neck incision postoperatively is one of the greatest benefits of SP-TART. The normal arm position conceals the surgical scar in the axilla, providing good cosmetic results [11]. Most of the patients undergoing thyroid surgery are females and are under the age of 40 years. In our data, the mean age of 100 patients was 41.8 years and 91 of them were female. Some young females, especially those who are interested in beauty, are reluctant to expose surgical wounds. Endocrine surgeons need to cater to those needs. The axillary approach has the advantage of avoiding swallowing symptoms postoperatively in addition to cosmetic outcomes [11]. Several studies revealed that TART is safe and feasible and not inferior to COT [19,20,21,22].

SP-TART is a safe operation regardless of the patient’s physique. Creating a working space via the axilla is easier in female patients with short height than in tall males or obese patients due to the difference in the extent of the skin flap [23]. However, this is not difficult for an experienced surgeon to overcome. Data from Yonsei University in South Korea revealed that obesity was not a major factor that affect the oncological and surgical outcomes of TART [24]. This study revealed that 6 patients have a body mass index of >30 kg/m^2^, and one of them experienced vocal cord palsy postoperatively due to intraoperative transection of the RLN. Making trans-axillary skin flaps is considered more difficult in male patients due to the long distance and tightness. Our data included 7 male patients, of whom two had postoperative complications. One experienced vocal cord palsy after TT and the other experienced transient hypoparathyroidism and chyle leakage after TT with mRND.

Generally, TART does not differ in the oncological outcome of low-risk groups with early thyroid cancer compared to COT [2,12,25,26,27]. The demand for TART increased as the number of patients with thyroid microcarcinoma increased in the 2010′s due to the development of ultrasound images in quality. Furthermore, TART was performed for advanced patients who needed mRND. The 5-year surgical and oncological outcomes between open mRND and robot mRND groups were not significantly different [28].

This study included three patients who underwent TT with ipsilateral mRND. Dissecting LNs at the upper jugular level was easier with SP robotic system than the previous robotic system developed for multi-port surgery. All the procedures, including both thyroid glands and supraclavicular LN dissection, were performed with the patient’s arm laterally extended. Afterward, de-docking and redocking were needed for more flap dissection to create a working space. The patient’s head was rotated toward the contralateral lobe and the external retractor [18] was reinserted so that the tip of the retractor could reach the upper jugular level of the ipsilateral lateral neck. We learned that additional retraction by an assistant was not necessary when dissecting the LNs at level II with the help of the flexible instruments and camera.

Studies on postoperative pain of TART were reported. Prospective studies reported that TART did not cause less postoperative pain than COT [13,29]. Similarly, we observed that quite a few patients had a hard time with postoperative pain at the location of subcutaneous flap dissection after SP-TART. We performed ultrasonography-guided pectoralis nerve block II before making an incision for some patients undergoing SP-TART (n = 28) and compared the visual analog scale with the no-block group (n = 27). The block group had significantly lower pain scores than the no-block group [30]. SP-TART may cause pain shortly postoperatively, but it can be overcome by a pectoralis nerve block.

One of the shortcomings of TART is sensory nerve injury caused by subcutaneous flap dissection [5]. Some patients appeal to paresthesia on the supra- and infra-clavicular region for several months postoperatively. The flexible SP robotic system enabled less extent of flap dissection which led to minimizing the sensory nerve injury.

This study has limitations. One is its retrospective nature. Additionally, there is something to consider as selection bias in this study. Getting enough profits from treatment covered by insurance is difficult in South Korea. Robotic surgery is not covered by public health insurance; thus, it has been facilitated for the last decade. Some patients can afford to pay private insurance, while others cannot. Therefore, there may be more demand for robotic surgery in certain regions with better socioeconomic status.

Of the total 100 cases in this study, 10 were complicated, and 5 of them were associated with the RLN and the parathyroid gland. Bleeding occurred in one case. No case was converted to open surgery intraoperatively, and there was no re-operation case postoperatively. Considering the small sample size of this study, SP-TART is safe and feasible with a low complication rate and short postoperative hospital stay. The complication rate will decrease as the surgical team gets more experience.

## 5. Conclusions

To our best knowledge, SP-TART is safe and technically feasible with a short incision length, a short hospital stay, and a relatively low complication rate. However, further prospective studies are needed to verify the technical feasibility and evaluate the operative outcomes. We expect this study to help endocrine surgeons who would like to perform SP-TART.

## Figures and Tables

**Table 1 medicina-58-01486-t001:** Baseline clinicopathological characteristics of the study patients.

Total 100 Cases	
Age (years)	41.8 ± 13.3 (range, 17–69)
Male:Female	1: 10.1 (9:91)
Body mass index (kg/m^2^)	23.4 ± 3.6 (range, 16.5–32.8)
Extent of operation	
LTT	81 (81.0%)
TT	16 (16.0%)
TT with mRND	3 (3.0%)
Direction of approach	
Right/Left	56 (56.0%)/44 (44.0%)
Pathologic subtype	
PTC	82 (82.0%)
Follicular adenoma	8 (8.0%)
NIFTP	3 (3.0%)
FTC	1 (1.0%)
Nodular Hashimoto thyroiditis	1 (1.0%)
Oncocytic (Hurthle cell) adenoma	1 (1.0%)
Nodular hyperplasia with oncocytic (Hurthle cell) changes	1 (1.0%)
PDTC	1 (1.0%)
Diffuse hyperplasia with Hashimoto thyroiditis	1 (1.0%)
No residual tumor (completion TT) *	1 (1.0%)
Tumor size (cm) **	
Sonographic (n = 98)	1.5 ± 1.3 (range, 0.3–7.0)
Pathologic (n = 97) ***	1.3 ± 1.2 (range, 0.1–5.9)
Thyroiditis	
Hashimoto/Focal	40 (40.0%)/6 (6.0%)

Data are expressed as the patient number (%) or mean ± SD. * The patient with PDTC underwent two surgeries to perform completion TT. ** The tumor size was measured based on the largest one when multifocal tumors were observed. *** The tumor size was not reported as measured in the pathology of one patient with PTC. Abbreviations: LTT, less than total thyroidectomy; TT, total thyroidectomy; mRND, modified radical neck dissection; PTC, papillary thyroid carcinoma; NIFTP, noninvasive follicular neoplasm with papillary-like nuclear features; FTC, follicular thyroid carcinoma; PDTC, poorly differentiated thyroid carcinoma.

**Table 2 medicina-58-01486-t002:** Perioperative outcomes according to the extent of operation.

	LTT (n = 81)	TT (n = 16)	TT with mRND (n = 3)
Operation time (min)	53.3 ± 13.7	86.3 ± 15.1	245.7 ± 36.7
Flap time	14.8 ± 2.8	22.8 ± 3.6	39.0 ± 5.6
Docking time	2.1 ± 0.9	2.0 ± 0.0	4.0 ± 0.0
Console time	22.5 ± 9.9	47.8 ± 12.2	158.0 ±14.4
Blood loss (ml)	8.4 ± 5.4	23.3 ± 48.9	38.3 ± 53.5
Hospital stay (POD)	2.0 ± 0.2	2.1 ± 0.3	3.7 ± 1.5
Postoperative complications	6 (7.4%)	2 (12.5%)	1 (33.3%)

Data are expressed as the patient number (%) or mean ± SD. Abbreviations: LTT, less than total thyroidectomy; TT, bilateral total thyroidectomy; mRND, modified radical neck dissection; POD, postoperative day.

**Table 3 medicina-58-01486-t003:** Clinicopathological characteristics of the patients with cancer.

Total 84 Cases	
Extent of operation	
LTT/TT/mRND	67 (79.8%)/14 (16.6%)/3 (3.6%)
Pathologic subtype	
PTC	82 (97.6%)
FTC	1 (1.2%)
PDTC	1 (1.2%)
Tumor size (cm)	
Sonographic	1.3 ± 1.3 (range, 0.3–7.0)
Pathologic	1.1 ± 1.2 (range, 0.1–5.9)
Multifocality	30 (35.7%)
ETE	
Minimal/Gross	50 (59.5%)/5 (6.0%)
Thyroiditis	
Hashimoto/Focal	36 (42.9%)/6 (7.1%)
LN dissection	81 (96.4%)
Harvested LNs	8.0 ± 9.0 (range, 0–56)
Positive LNs	2.4 ± 4.8 (range, 0–27)
T stage	
T1a/T1b/T2/T3a/T3b	59 (70.2%)/12 (14.2%)/3 (3.6%)/5 (6.0%)/5 (6.0%)
N stage	
N0/N1a/N1b/Nx	38 (45.2%)/39 (46.4%)/3 (3.6%)/4 (4.8%)
TNM stage	
Stage I/II	79 (94.0%)/5 (6.0%)

Data are expressed as the patient number (%) or mean ± SD. Abbreviations: LTT, less than total thyroidectomy; TT, bilateral total thyroidectomy; mRND, modified radical neck dissection; PTC, papillary thyroid carcinoma; FTC, follicular thyroid carcinoma; PDTC, poorly differentiated thyroid carcinoma; ETE, extrathyroidal extension; LN, lymph node; T, tumor; N, node; M, metastasis; TNM, tumor-node-metastasis.

**Table 4 medicina-58-01486-t004:** Postoperative complications of the study participants.

Total 100 Cases	
Vocal cord palsy	3 (3.0%)
Surgical site infection	2 (2.0%)
Transient hypoparathyroidism *	2 (2.0%)
Chyle leakage **	1 (1.0%)
Drain insertion site bleeding	1 (1.0%)
Wound dehiscence	1 (1.0%)

Data are expressed as the patient number (%). * One was developed after TT while the other was after TT with mRND. ** The same patient experienced chyle leakage and transient hypoparathyroidism after TT with mRND, left. Abbreviations: TT, bilateral total thyroidectomy; mRND, modified radical neck dissection.

## Data Availability

The data underlying this article cannot be shared publicly to maintain the privacy of individuals that participated in the study. The data will be shared upon reasonable request to the corresponding author.

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
