# Peer review of "Safety and Feasibility of Single-Port Trans-Axillary Robotic Thyroidectomy: Experience through Consecutive 100 Cases"

_medicina, 2022, doi:10.3390/medicina58101486_

Round 1

Reviewer 1 Report

The article is very well organized. 

I would strongly suggest to the authors introduce in the introduction section chemicals as a potential role in the etiology and pathogenesis of thyroid carcinoma. It would be nice to hear how often these types of data are involved in an anamnesis. That would give additional value to the manuscript. 

Reviewer 2 Report

I have the following observations:

1.  The title of the manuscript is appropriate for the manuscript.

2.  The Abstract is an appropriate summary of the entire manuscript

3.  The Keywords are adequate.

4.  In the Introduction, section author described in line 41 as TART is better than COT and he contradicted his statement in lines 42 & 43. That needs to be checked again.

5.  The methodology is appropriately described however needs some clarification in different sections.

.   Patients: the author has not mentioned the selection criteria of their patients for the SP-TART, which will helpful for the readers to select their patients in the future.

b.  Operative procedure: an author needs to mention in the SP-TART technique, how they created the flap before docking the SP, as his technique may be different than the other robotic thyroidectomies.

c.    Author needs to describe in detail how they applied the retractor before putting the SP or have they used the CO2 insufflation during the surgery. It will help for future reproducibility

6.  In the Result section, Table -1, the author has mentioned thyroiditis in 40/6, however, he has mentioned two other patients with Nodular Hashimoto thyroiditis & Diffuse hyperplasia with Hashimoto thyroiditis, one in each category.

7.  In the Postoperative complication section the author needs to clarify the complications in relation to extent of surgery or relation to the pathology, it will be helpful for future readers.

8.  The discussion section elaborated appropriately with the current literature.

9.  Conclusion is appropriate and matching with the manuscript's objectives.

10.       References are appropriately mentioned in the manuscript
